# Genetic Analyses of Tanzanian Local Chicken Ecotypes Challenged with Newcastle Disease Virus

**DOI:** 10.3390/genes10070546

**Published:** 2019-07-17

**Authors:** Muhammed Walugembe, James R. Mushi, Esinam N. Amuzu-Aweh, Gaspar H. Chiwanga, Peter L. Msoffe, Ying Wang, Perot Saelao, Terra Kelly, Rodrigo A. Gallardo, Huaijun Zhou, Susan J. Lamont, Amandus P. Muhairwa, Jack C.M. Dekkers

**Affiliations:** 1Department of Animal Science, Iowa State University, 2255 Kildee Hall, Ames, IA 50011, USA; 2Department of Veterinary Medicine and Public Health, Sokoine University, P.O. Box 3000 Chuo Kikuu, Morogoro, Tanzania; 3Department of Animal Science, University of Ghana, P.O. Box LG 25 Legon, Accra, Ghana; 4Department of Animal Science, University of California, Davis, CA 95616, USA; 5School of Veterinary Medicine, University of California, Davis, CA 95616, USA

**Keywords:** NDV, GWAS, Tanzanian local ecotypes, QTL, immune response

## Abstract

Newcastle Disease (ND) is a continuing global threat to domestic poultry, especially in developing countries, where severe outbreaks of velogenic ND virus (NDV) often cause major economic losses to households. Local chickens are of great importance to rural family livelihoods through provision of high-quality protein. To investigate the genetic basis of host response to NDV, three popular Tanzanian chicken ecotypes (regional populations) were challenged with a lentogenic (vaccine) strain of NDV at 28 days of age. Various host response phenotypes, including anti-NDV antibody levels (pre-infection and 10 days post-infection, dpi), and viral load (2 and 6 dpi) were measured, in addition to growth rate. We estimated genetic parameters and conducted genome-wide association study analyses by genotyping 1399 chickens using the Affymetrix 600K chicken SNP chip. Estimates of heritability of the evaluated traits were moderate (0.18–0.35). Five quantitative trait loci (QTL) associated with growth and/or response to NDV were identified by single-SNP analyses, with some regions explaining ≥1% of genetic variance based on the Bayes-B method. Immune related genes, such as ETS1, TIRAP, and KIRREL3, were located in regions associated with viral load at 6 dpi. The moderate estimates of heritability and identified QTL indicate that NDV response traits may be improved through selective breeding of chickens to enhance increased NDV resistance and vaccine efficacy in Tanzanian local ecotypes.

## 1. Introduction

Newcastle disease (ND) is a major threat to poultry globally, and severe outbreaks of the velogenic strains of Newcastle disease virus (NDV) in African local chicken populations often have devastating economic impacts on households. In Africa, local chicken production is characterized by low input production systems [1,2] and serves as a major source of high quality protein (eggs and meat) and financial assets, particularly for children and women [2,3,4]. Because local chickens are managed in backyards or under a free-range scavenging system [2], households often lose entire chicken flocks to poultry diseases such as NDV [5,6]. Farmers usually control NDV through vaccination, but this is an inadequate control method in many small-holder farms in rural Africa because of limited husbandry and biosecurity practices [2], high costs of vaccines given that commercial formulations are only available in large volumes, difficulty in administering vaccines, and instability of vaccines because of lack of a “cold chain” [7].

Newcastle Disease Virus is an avian paramyxovirus type 1 (APMV-1) virus that belongs to the family *Paramyxoviridae* and genus *Avulavirus* and is a devastating poultry pathogen [8]. Chickens infected with NDV can show various clinical signs that vary with the type of viral strain [9]. NDV strains are categorized into three pathological groups: lentogenic, mesogenic, and velogenic [10,11]. Lentogenic strains can cause subclinical infections or mild enteric or respiratory disease and include the LaSota strains that are often used for vaccination [12]. Mesogenic strains usually cause respiratory and neurological disease, with low mortality, particularly in young chickens. Velogenic strains are characterized by systematic infections with clinical signs that include hemorrhagic lesions in the gastro-intestinal tract, diarrhea, severe respiratory disease, neurologic disease, and sudden death [13].

African local chicken ecotypes/breeds are characterized by production under limited management of low feed supply and are well adapted to harsh climatic conditions, as evidenced by the presence of selection signatures across their genomes [14,15]. However, entire chicken flocks are often lost in the case of velogenic ND outbreaks. This is partly attributed to the absence of local chicken populations that are resilient to NDV. In addition, there is lack of organized breeding programs for local chicken ecotypes with improved NDV resilience. Several studies conducted in developed countries have revealed genomic regions associated with NDV response traits in broilers, intercross lines [16,17], and egg-laying chickens [18,19]. However, there is limited literature addressing the genetic basis of various NDV response traits in African chicken populations. Some studies have reported genetic parameters for some traits [20,21], but comprehensive genome-wide studies to understand the genetic basis of NDV response traits are lacking. On the basis of genomic information discovered in such comprehensive studies, a selective breeding program could be developed to produce local chicken populations that could perform favorably in challenging environmental conditions where NDV is endemic. The objective of our study was to estimate genetic parameters and use genome-wide association studies (GWAS) to identify quantitative trait loci (QTLs) associated with NDV response traits in Tanzanian local chicken ecotypes using an available high-density single nucleotide polymorphism (SNP) chip.

## 2. Materials and Methods

### 2.1. Experimental Design

All animal procedures were approved by the Institutional Animal Care and Use Committee of University of California, Davis (#:17853)

Breeder chickens consisted of 65 sires and 324 dams from three popular Tanzanian chicken ecotypes, Ching’wekwe (Ching), Kuchi, and Morogoro Medium (MoroMid) that were randomly collected from the five regions of Tanga, Shinyanga, Morogoro, Singida, and Mwaza, representing the central, coastal, and lake zones of Tanzania [22]. Sires and dams were vaccinated using the recommended vaccination schedule by project veterinary personnel. Challenge experiments were conducted for a total of 1399 chicks (Ching (477), Kuchi (315), and MoroMid (607)) from hatch to 38 days of age (doa) across five replicates. All birds were raised under similar conditions with *ad libitum* access to feed and water.

Blood samples were collected from all birds at 27 doa and IDEXX NDV ELISA was used to quantify maternal antibody levels (IDEXX Laboratories, Inc., Westbrook, ME, USA). At 28 doa, all birds were challenged via the oculo-nasal route with 10^7^, 50% embryonic infectious dose (10^7^EID_50_) of a live attenuated type B1 LaSota lentogenic NDV strain. This lentogenic vaccine strain induces an immune response without causing severe disease and death, allowing for observation of phenotypic responses to the virus. To quantify viral load, viral RNA was isolated from lachrymal fluid samples at 2 and 6 days post infection (dpi) and quantified using qPCR, as described by Rowland et al. [18]. The mean viral RNA was computed per sample and transformed to log_10_ for downstream analyses. Viral clearance was computed as the difference between viral loads at the two time points divided by viral load at 2 dpi. Blood samples were collected and ELISA was used to measure the anti-NDV antibody levels at 10 dpi, which is the time required for birds to generate an acquired immune response [18,23]. The anti-NDV antibody level data were also transformed to log_10_. Body weights were measured at hatch and at 7, 14, 21, 28 (0 dpi), 34 (6 dpi), and 38 (10 dpi) doa. Pre- and post-infection growth rates were calculated as grams per day from these weights using linear regression of weight on age. Outliers greater than three standard deviations from the mean were removed for all response traits.

### 2.2. Genotyping and Quality Control

Blood samples were collected from chicks before challenge using Whatman FTA cards (Sigma-Aldrich, St. Louis, MO, United States). Genomic DNA was isolated from the FTA cards and genotyping was conducted using the Affymetrix Axiom^®^ 600k Array at GeneSeek (Lincoln, NE, USA). Genotyping Array annotation files were based on the chicken *Gallus gallus* genome version 5 (Thermo Fisher Scientific Inc., Calsbad, CA, USA). Genotype data quality filtering was performed with Axiom^™^ Analysis Suite 3.1 (Applied Biosystems, Thermo Fisher Scientific Inc., Calsbad, CA, USA) and included single nucleotide polymorphism (SNP) call rate ≥99% and minor allele frequency ≥0.05. Other Affymetrix genotype metrics used for filtering, with their corresponding cutoffs, are listed in Table 1. After filtering, a total of 396,055 SNPs remained. Imputation of missing genotypes was performed using Fimpute [24].

### 2.3. Population Stratification

Population structure was examined by constructing a Multi-dimensional Scaling (MDS) plot in two dimensions using the cluster algorithm in the PLINK v1.9 software [25]. Shared ancestry of birds was explored using the Admixture software [26], with the number of subpopulations ranging from 1 to 4. The optimal number of subpopulations was determined based on the lowest cross-validation error rate and was determined to be 2. The generated population proportions for each individual were used in downstream GWAS analyses.

### 2.4. Genetic Parameters and Correlations

Estimation of variance components and heritabilities was done using ASReml 4 [27] based on the following univariate mixed linear animal model:
(1)$$Y_{ijlkmn}=\mu +D_{i}+R_{j}+S_{l}+C_{k}+A_{m}+X_{n}+e_{ijlkmn}$$
where *Y* is the dependent phenotype variable: pre- and post-infection growth rate, antibody at 10 dpi, viral load at 2 dpi, and viral load at 6 dpi. Fixed effects included death prior to 10 dpi (*D* = 0/1), trial (replicate, *R* = 1–5), and sex (*S* = male/female). Only one covariate, population proportion (*C*), obtained as described above, was fitted. Random effects included animal genetic effects (*A*), using the genomic relationship matrix computed based on procedures reported by [28], *X* to account for maternal effects, and residuals (*e*). The dam effect (*X*) was removed for some traits because it was not significant. For viral load at 2 and 6 dpi, and antibody at 10 dpi, qPCR plate (55 and 60 plates at 2 and 6 dpi, respectively) and replicate plate (46), respectively, were added as fixed effects. Phenotypic variance was obtained by summation of variance due to animal genetic, dam, and residuals. Heritability was computed as a ratio of the estimates of animal genetic to phenotypic variance. Bivariate animal models were used to estimate pairwise genetic and phenotypic correlations between traits, with the same fixed and random effects as specified for the univariate analyses.

### 2.5. Genome-Wide Association Analyses

Two approaches were used for whole genome association analyses. First, a Bayesian approach called Bayes B [29], as implemented in the Gensel software [30], was used to compute the genetic variance accounted for by every-one mega base (Mb) window of SNPs. Model (1) above was used for the analyses but with A was replaced by SNP genotype effects. The genotypes at all SNPs, coded as −10/0/10 for AA/AB/BB, respectively, were fit in a Bayes B approach using a prior probability of the SNP having no effect (pi) of 0.999. In total, 41,000 iterations of a Markov chain were run for each trait, with 1000 iterations burn-in and 100 iterations as the output frequency. We used Bayes B instead of Bayes C approach because it performs better in detecting QTL windows compared to Bayes C.

For the second approach, single SNP association analyses were conducted using the R package GenABEL [31], with a hierarchical generalized linear model [32]. The same fixed (class) and covariate effects described in the Gensel analyses were used in the GenABEL analyses. A polygenic model was fitted using the “*polygenic_hglm*” function, with a genomic kinship matrix that was created using the *ibs()* function. To test for association between a trait and genotypes at a SNP for related individuals, the “*mmscore*” function was used and residuals were obtained from the *polygenic_hglm* analysis. The *mmscore* function was performed using the formula described by Rowland et al. [18].

### 2.6. Multiple Test Correction

To determine significance thresholds for the GenABEL analyses, multiple test correction was performed. Suggestive genome-wide significance thresholds of 10 and 20% were computed using a modified Bonferroni correction as 0.1 or 0.2 divided by the number of independent tests. To determine the number of independent tests, the SNP genotype data were separated by chromosome and then further divided to form chromosomal segments such that the number of SNPs were equal to half the number of animals, as described by Waide et al. [33]. The number of independent tests in each segment was determined by the minimum number of principle components required to account for 95% of the variance among genotypes in each segment. The total number of independent tests was the sum across segments.

### 2.7. Bioinformatics Analyses

Gene annotation for 1-Mb windows that explained more than 1% of genetic variance was obtained using NCBI’s Genome Data Viewer on the chicken genome version Gallus gallus 5 (https://www.ncbi.nlm.nih.gov/genome/gdv/)

## 3. Results

### 3.1. Population Stratification and Phenotypic Data

Population stratification using Admixture and Multi-Dimensional Scaling showed overlaps among the three chicken ecotypes (Figure 1 and Figure 2). The clustering analyses indicated that ecotypes assigned to birds showed shared ancestry of genotypes among birds across ecotypes (Figure 1). Two main clusters were identified, with Ching and MoroMid belonging to one discrete cluster and Kuchi to the other cluster. Admixture analyses based on identity by state also showed clear overlaps between the three ecotypes. Birds mainly belonged to two populations, with Ching and MoroMid having higher average proportions of population one (0.78 and 0.75 for Ching and MoroMid, respectively), while Kuchi birds had a higher (0.67) average proportion of population two.

In total, 1399 chicks were challenged with NDV. The mean ± standard error for growth rate was 5.12 ± 1.31 and 6.85 ± 2.82 g/d for pre- and post-infection growth rates, respectively (Table 2). The mean log_10_ anti-NDV antibody level was 3.45 ± 0.45. The mean log_10_ viral copy number was 4.72 ± 1.03 and 4.25 ± 1.18 for 2 and 6 dpi, respectively. Mean viral clearance from 26 dpi was 6%.

### 3.2. Genetic Parameter Estimates

Estimates of heritability from single-trait analyses are shown in Table 2. Heritability estimates from single-trait and bivariate analyses were similar. Estimates of heritability for pre- and post-infection growth rates were moderate, 0.35 and 0.21, respectively. Viral load was moderately heritable, with estimates of 0.18 and 0.35 at 2 and 6 dpi, respectively, but the estimate of heritability for viral clearance was not different from zero. The heritability estimate for anti-NDV antibody level (10 dpi) was 0.22.

Estimates of phenotypic and genetic correlations among the traits are shown in Table 3. For phenotypic correlations, pre- and post-infection growth rates were highly correlated (0.54 ± 0.02) and both were positively correlated with antibody levels and negatively correlated with viral load at 2 and 6 dpi. Anti-NDV antibody levels were positively correlated with viral load at 2 and 6 dpi. Viral clearance was positively and negatively correlated with viral load at 2 and 6 dpi, with estimates of 0.18 ± 0.03 and –0.29 ± 0.03, respectively.

Pre- and post-infection growth rates were genetically highly correlated (0.74 ± 0.08) (Table 3). Viral clearance was genetically positively correlated with pre- and post-infection growth rates and with antibody level; genetically, birds with higher antibody levels at 2 dpi had higher viral clearance. Viral clearance was negatively correlated with viral load at 6 dpi but not significantly (−0.11 ± 0.21). Viral load at 6 dpi was negatively correlated with pre and post-infection growth rate with estimates of −0.23 and −0.13, respectively.

### 3.3. Genome-Wide Association Studies

A total of 1399 animals and 396,055 SNPs remained after quality control and were utilized for GWAS analyses. Because the chicken ecotypes were highly heterogeneous and admixed, analyses were performed across ecotypes by fitting population proportion (C) to account for the structure of the populations. Using principle component analysis, 71,374 components accounted for 95% of the variance of SNP genotypes across the genome. Bonferroni corrected thresholds of 10 (*p*-value = 2.05 × 10^−6^) and 20% (*p*-value = 7.035 × 10^−6^) were used to declare suggestive associations in the single SNP GenABEL analyses. Markers significantly associated with pre- and post-infection growth rates, antibody, and viral load at 2 and 6 dpi from single-SNP GenABEL analyses are presented in Table 4. Gensel results (≥0.5% genetic variance in a 1-Mb window) obtained using the Bayes-B method are presented in Table 5.

There were 10, 1-Mb windows on eight chromosomes that together explained 6.5% of genetic variance for pre-infection growth rate. The location of two suggestive significant SNPs in two QTL on chromosomes 3 and 22 (Figure 3) corresponded with two 1-Mb windows identified for pre-infection growth rate. For post-infection growth rate, 2 windows on two chromosomes explained 1.7% of genetic variance. The window that explained the most genetic variance (1.2%) corresponded with the location for a suggestive significant SNP on chromosome 19 (Figure 4). One SNP on chromosome 2 (Figure 5) and one window explaining > 1% genetic variance on chromosome 9 were associated with anti-NDV antibody levels. None of the 1-Mb windows corresponded in location with the significant SNP on chromosome 2 from single-SNP analyses. Four windows on three chromosomes explained a total of 3.8% genetic variance for viral load at 2 dpi. One significant SNP on chromosome 5 (Figure 6) corresponded in location with a 1-Mb window explaining 2% of genetic variance. Three windows on three chromosomes together explained 13.7% of genetic variance for viral load at 6 dpi. Three SNPs associated with viral load at 6 dpi (Figure 7B) corresponded in location with a 1-Mb window that explained 12.4% of genetic variance in the Bayes-B analysis (Table 4). Although heritability was low for viral clearance (Table 2), five windows explaining >1% genetic variance were identified. Four significant SNP locations were in correspondence with these windows (Figure 8).

## 4. Discussion

### 4.1. Population Stratification

The admixture results for the three chicken ecotypes (Kuchi, Ching and MoroMid) indicated a common genetic background of the Tanzanian local chicken ecotypes that were evaluated. This admixture can be attributed to inter-mating (breeding) among chickens across different areas of the country and movement of chickens between country districts through trading and market chains. The MoroMid and Ching ecotypes were sampled from districts (regions) that neighbor each other, compared to Kuchi sample districts. The complimentary Admixture (Figure 1) and MDS (Figure 2) plots support this geography, where Ching and MoroMid birds had similar admixture patterns and clustered together in the MDS plot compared to Kuchi.

### 4.2. Genetic Parameters

Estimates of heritability were moderate for most traits, ranging from 0.18 for viral load at 2 dpi to 0.35 for pre-infection growth rate. Viral clearance had the lowest heritability estimate, at 0.04. These results agree with previous findings from commercial egg laying chickens that were challenged with the LaSota NDV strain [18,19]. Heritability estimates for antibody level at 10 dpi were similar to those for antibody level in two Tanzanian (Kuchi and MoroMid chicks) ecotypes measured at 2 weeks post-infection [20]. To the best of our knowledge, this study is the first to report heritability estimates for growth rates and viral load in local Tanzanian chicken ecotypes. The moderate estimates of heritability suggest that all NDV response traits measured could be improved through selective breeding of chickens to enhance NDV resilience.

Growth rates were negatively and positively genetically correlated with viral load at 2 and 6 dpi, and viral clearance, respectively. Although these correlations were low, this indicates that selection for decreased viral load at 6 dpi and increased viral clearance could increase pre- and post-infection growth rates. In addition, the positive genetic correlations between anti-NDV antibody levels and growth rates indicate that selection for increased antibody levels would also increase growth rate. Our results contrast with the findings of Rowland et al. [18] in commercial layers, which showed unfavorable genetic correlations of viral load at 6 dpi and antibody levels with pre- and post-infection growth rates. The estimates of genetic correlations of anti-NDV antibody levels with viral load at 2 and 6 dpi were not different from zero. This suggests that 10 dpi anti-NDV antibody may not be a good indicator trait for the ability of the virus to replicate within the host. Although we identified moderate genetic correlations between some of the traits, these estimates should be considered with care because of the high standard errors.

### 4.3. Genome-Wide Association Analyses

Two analysis platforms, Gensel and GenABEL, were used for GWAS analyses. We utilized the Bayes B approach in the Gensel software to compute genetic variance accounted for by a 1-Mb window for the NDV response traits. The Gensel software does not allow for fitting random effects in the model, except SNP effects. An R package, GenABEL, was used to identify individual SNPs associated with the NDV response traits. GenABEL allows fitting single SNPs one-by-one during the association analyses.

One suggestive QTL region was identified for post-infection growth rate based on a significantly associated SNP on chromosome 19 at 1.6 Mb. The location of the QTL identified by GenABEL analyses corresponded with a 1-Mb window (Table 4) that explained 1.2% of genetic variance based on the Gensel analyses. This SNP is within the intron of the AUTS2 activator of transcription and developmental regulator gene. The role of AUST2 is not well known in chickens but previous studies in other species found AUST2 to be associated with various neurological diseases [34,35]. A study conducted in zebrafish reported an increase in cell proliferation in the brain when AUST2 was knocked down [34]. Therefore, this gene could be vital for growth under NDV challenge conditions. Another gene, SBDS ribosome maturation factor, was 793,675 bp downstream of the significant SNP. In mammalian cells, SBDS has been implicated in several pathways, including cell motility [36], regulation of reactive oxygen species, and ribosome biogenesis [37]. The SBDS gene was differentially expressed at 14 dpi in White Leghorn chickens inoculated with *Salmonella enterica* serovar Enteritidis. The SBDS gene could be important in the regulation of cellular processes during disease challenge in chickens.

Single-SNP analyses revealed one significant suggestive SNP associated with antibody level. A window explaining >1% of genetic variance for antibody level contained 16 genes (Table 6). However, the location of this window did not correspond with the significant SNP from single SNP analyses. One of these genes from this window, HES1, is a critical mediator of canonical notch signaling in lymphocyte development and transformation in mice [38]. HES1 is expressed in B, T, and STEM cell blood lineages, and is important in transmitting Notch functions [39,40].

One significant suggestive QTL for viral load at 2 dpi was identified on chromosome 5 by the GenABEL analyses. This SNP corresponds in location with a 1-Mb window that explained 2% of genetic variance based on the Gensel analyses. The significant SNP was within the intron of the PLEKHH1 gene, which interacts with the MYC transcription factor to activate the transcription of growth-related genes [41]. These results indicate that this SNP could be the first to associate the PLEKHH1 gene with viral load at 2 dpi. Other genes located in this 1-Mb window (Table 6) are reported to be important in immune response, including ZFP36L1, SLC39A9, and ACTN1. ZFP36L1 is an important regulator of innate immune response and may modulate *Mkp-1* mRNA basal levels to control p38 MAPK activity during lipopolysaccharide stimulation [42]. SLC39A9 induces an increase in extracellular zinc levels that leads to Akt and Erk activation/phosphorylation in response to B cell-receptor activation [43]. SLC39A9 may be an important gene in the early stages of NDV infection and may play an important role in the chickens’ adaptive immune response. Another gene, ACTN1 has been connected to phagocytosis and the immune system [44].

The QTL on chromosome 24 associated with viral load at 6 dpi had the most significant and largest number of SNPs identified in our study. The approximate 1-Mb window that contained these SNPs also had the highest genetic variance explained and contained 36 genes (Table 6). Some of these genes, including TIRAP, ETS1, KIRREL3, and ST3GAL4, are potential candidates for this QTL. TIRAP had three significant SNPs downstream of it that were associated with viral load at 6 dpi (Table 5). TIRAP is a Toll-interleukin 1 receptor [45] that recognizes pathogens within a host and is part of the microbial pathogen recognition Toll-like receptor system [46]. TIRAP is an adaptor protein that activates TLR4 signaling and is involved in modulating early innate immune response through detection of viral envelope glycoprotein [47]. Another immune response gene in the identified QTL region was ETS1, which is a transcription factor that regulates cytokines and chemokine gene expression [48] and is required for development of natural killer cells [49]. ETS1 may be an important gene in the early stages of NDV infection and could activate the chickens’ adaptive immune response. A previous study [19] conducted in commercial layer chickens identified significant SNPs in the same region as the current study for viral load at 6 dpi. That study investigated resistance to NDV of Hy-Line Brown chickens under the effects of heat stress, and authors utilized the GenABEL analysis R package for their association analyses. The two analysis platforms (GenABEL and Gensel) utilized in the current study revealed the same potential candidate genes on chromosome 24 associated with viral load at 6 dpi as identified by Saelao et al. [19]. A parallel study to Saelao et al. [19] investigated resistance to NDV of the same bird population, but without heat stress [18]. That study did not identify any significant SNPs associated to viral load at 6 dpi. A possible reason why our results for chromosome 24 agree with Saelao et al. [19] but not with Rowland et al. [18] could be because Tanzanian local chicken ecotypes in our study were exposed to natural heat stress. Experiments were conducted during the hottest months of January to May, and NDV challenge during this period may have influenced our results.

## 5. Conclusions

Our results revealed that heritability estimates for most NDV response traits were moderately high. Growth rates were positively genetically correlated with anti-NDV antibody level and were negatively correlated with viral load at 6 dpi. The genetic correlation between anti-NDV antibody levels and viral load at 2 and 6 dpi were not different from zero. Six suggestive QTL for NDV response traits were identified, some of which were consistent with QTL identified in previous independent studies. The strongest QTL region was identified on chromosome 24 and contained several candidate genes, including EST1, TIRAP, and KIRREL3, that could be vital in the chickens’ response to NDV infection. The moderate estimates of heritability and the identified QTL for NDV response traits, suggest that NDV response traits can be improved through selective breeding of chickens to enhance NDV resistance and vaccine efficacy of chickens in regions where NDV is endemic. The variants and important genomic regions identified in this study warrant further investigations to comprehensively understand underlying molecular mechanisms of NDV challenge.

## Figures and Tables

**Figure 1 genes-10-00546-f001:**
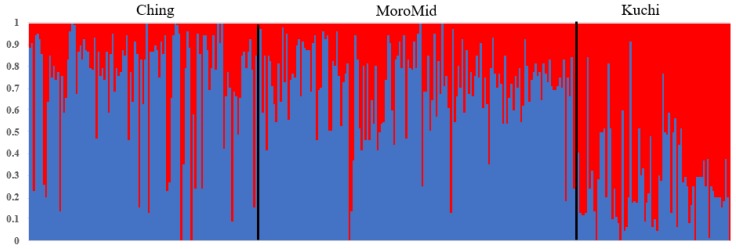
The admixture plot showing mixed ancestry among individuals for the three populations. ecotypes; Ching, Kuchi and MoroMid.

**Figure 2 genes-10-00546-f002:**
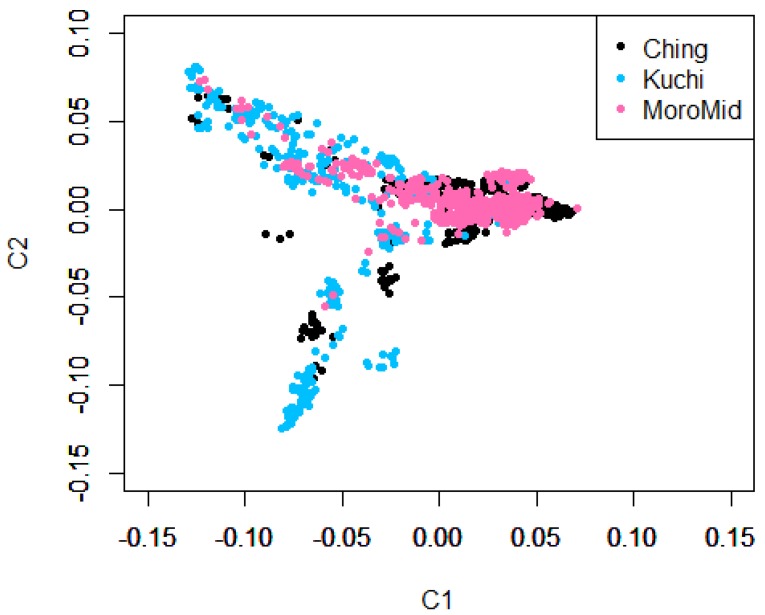
Multi-dimensional scaling (MDS) plot showing the three sampled population ecotypes.

**Figure 3 genes-10-00546-f003:**
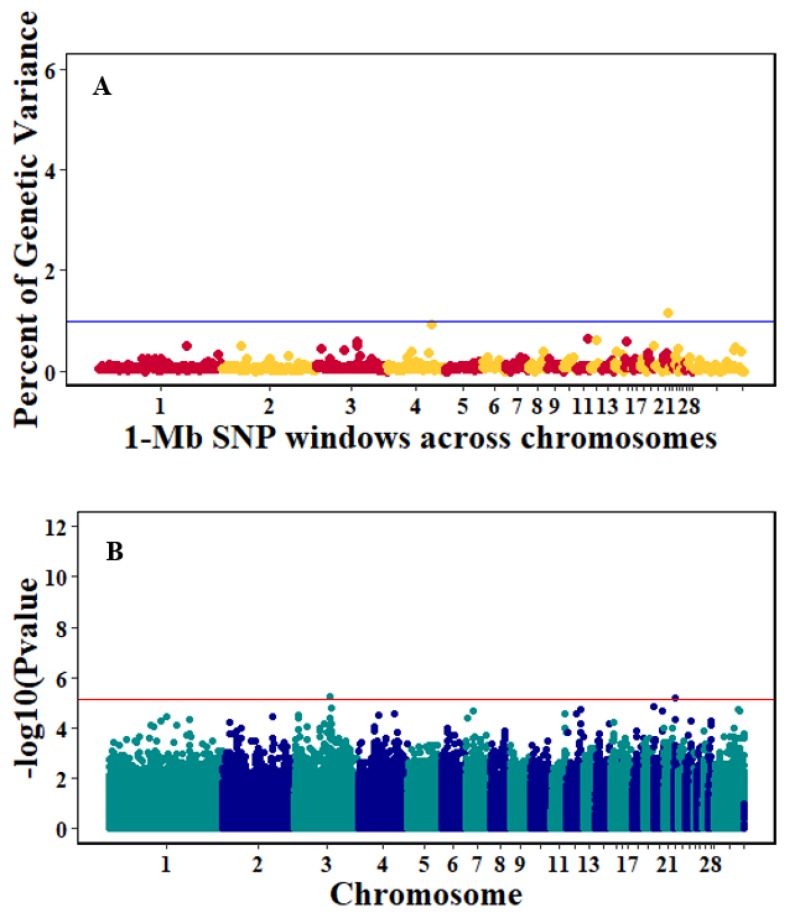
Manhattan plots showing the genome-wide association results for pre-infection growth rate using Bayes-B and single-SNP analyses. (**A**) Bayes-B results show the percent of genetic variance explained by 1-Megabase (1-Mb) non-overlapping window of SNPs across chromosomes. (**B**) Single-SNP results show the −log_10_(*p*-value) of ordered SNPs across the chromosomes. The blue and red lines indicate genome-wide significance at 1% genetic variance and 20% genome-wise significance.

**Figure 4 genes-10-00546-f004:**
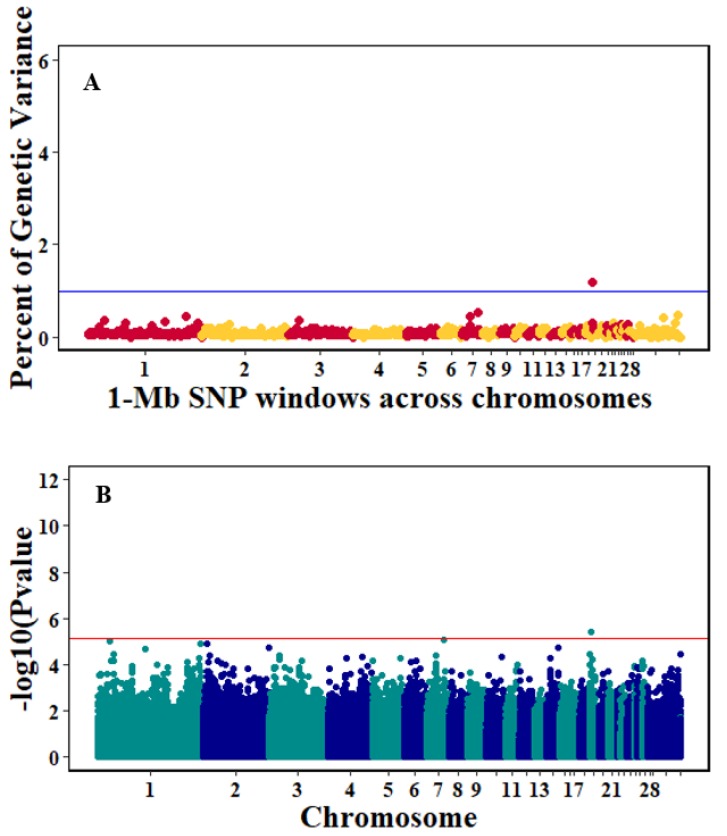
Manhattan plots showing the genome-wide association results for post-infection growth rate using Bayes-B and single-SNP analyses. (**A**) Bayes-B results show the percent of genetic variance explained by 1-Megabase (1-Mb) nonoverlapping windows of SNPs across chromosomes. (**B**) Single-SNP results show the −log_10_(*p*-value) of ordered SNPs across the chromosomes. The blue and reds lines indicate genome-wide significance at 1% genetic variance and 20% suggestive adjusted Bonferroni correction, respectively.

**Figure 5 genes-10-00546-f005:**
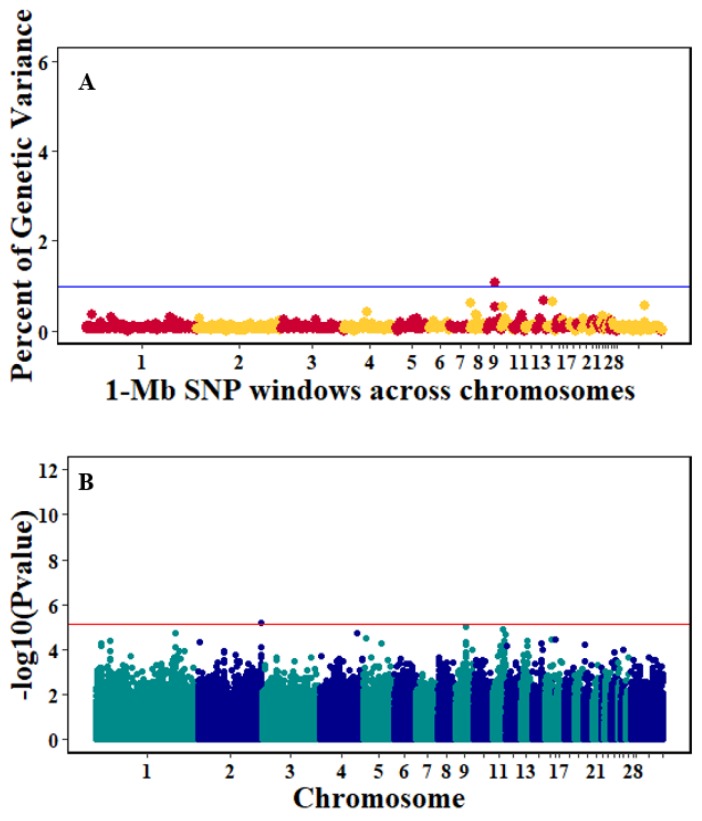
Manhattan plots showing the genome-wide association results for antibody levels at 10 dpi using Bayes-B and single-SNP analyses. (**A**) Bayes-B results show the percent of genetic variance explained by 1-Megabase (1-Mb) nonoverlapping windows of SNPs across chromosomes. (**B**) Single-SNP results show the −log_10_(*p*-value) of ordered SNPs across the chromosomes. The blue and red lines indicate genome-wide significance at 1% genetic variance and 20% suggestive adjusted Bonferroni correction, respectively.

**Figure 6 genes-10-00546-f006:**
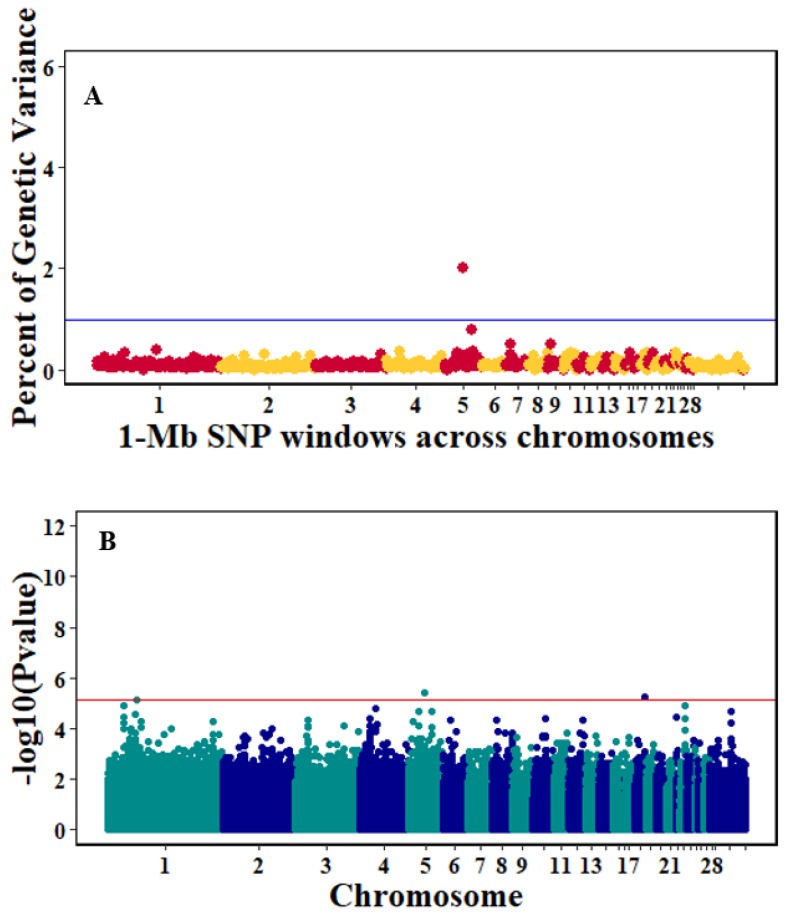
Manhattan plots showing the genome-wide association results for viral load at 2 dpi using Bayes-B and single-SNP analyses. (**A**) Bayes-B results show the percent of genetic variance explained by 1-Megabase (1-Mb) nonoverlapping windows of SNPs across chromosomes. (**B**) Single-SNP results show the −log_10_(*p*-value) of ordered SNPs across the chromosomes. The blue and red lines indicate genome-wide significance at 1% genetic variance and 20% suggestive adjusted Bonferroni correction, respectively.

**Figure 7 genes-10-00546-f007:**
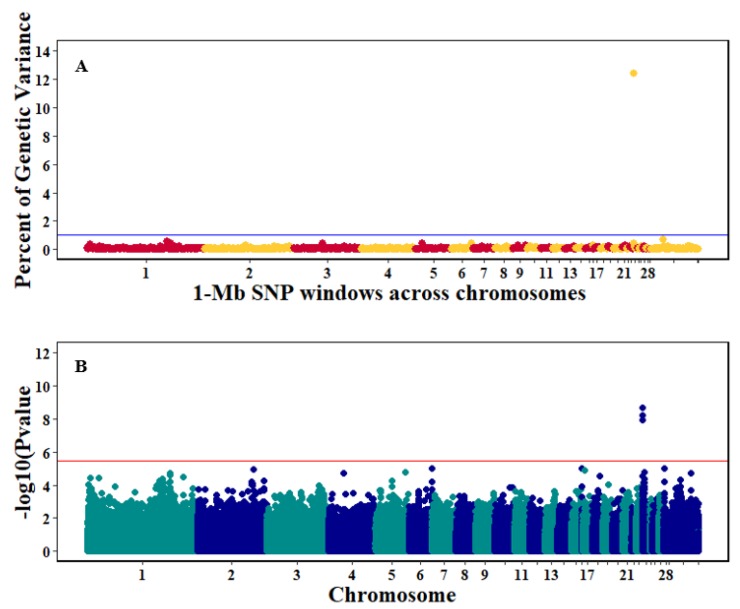
Manhattan plots showing the genome-wide association results for viral load at 6 dpi using Bayes-B and single-SNP analyses. (**A**) Bayes-B results show the percent of genetic variance explained by 1-Megabase (1-Mb) nonoverlapping windows of SNPs across chromosomes. (**B**) Single-SNP results show the −log_10_(*p*-value) of ordered SNPs across the chromosomes. The blue and red lines indicate genome-wide significance at 1% genetic variance and 20% suggestive adjusted Bonferroni correction, respectively.

**Figure 8 genes-10-00546-f008:**
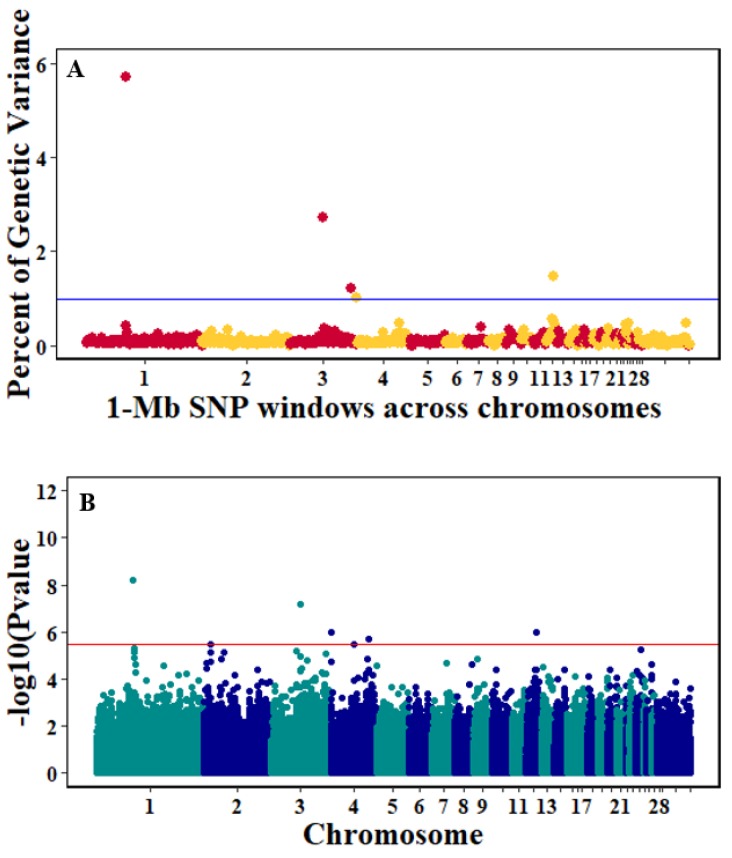
Manhattan plots showing the genome-wide association results for viral clearance using Bayes-B and single-SNP analyses. (**A**) Bayes-B results show the percent of genetic variance explained by 1-Megabase (1-Mb) nonoverlapping windows of SNPs across chromosomes. (**B**) Single-SNP results show the −log_10_(*p*-value) of ordered SNPs across the chromosomes. The blue and red lines indicate genome-wide significance at 1% genetic variance and 20% suggestive adjusted Bonferroni correction, respectively.

**Table 1 genes-10-00546-t001:** Genotype quality metrics provided by Affymetrix and the requirements used in quality control filtering.

Affymetrix Genotype Metric	Metric Description	Requirement
Nclus	Number of genotype clusters	≥2
CR	% of samples with genotype call other than "No call" for SNP	≥99%
MinorAlleleFrequency	Min (PA, PB), where PA is frequency of Allele A, and PB = 1−PA	≥0.05
FLD	Measure of the cluster quality of a probeset	≥5.12
HomRO	Distance to zero in the Contrast dimension (X position) from the center of the homozygous cluster that is closest to zero	≥0.47
HomFLD	Version of FLD computed for the homozygous genotype clusters	≥13.34
HetSO	Measures how far the heterozygous cluster center sits above the homozygous cluster centers in the Size dimension (Y position)	≥−0.35
ConversionType	Probeset classification	≠OTV
BB.varX	Contrast (X position) variance for BB cluster	≤0.85
BB.varY	Size (Y position) variance for BB cluster	≤0.69
AB.varX	Contrast (X position) variance for AB cluster	≤0.75
AB.varY	Size (Y position) variance for AB cluster	≤0.78
AA.varX	Contrast (X position) variance for AA cluster	≤0.79
AA.varY	Size (Y position) variance for AA cluster	≤0.51

**Table 2 genes-10-00546-t002:** Trait statistics and estimates (±SE) of variance components from univariate analyses.

Trait	N^3^	Mean^4^	SD^5^	Heritability + SE	Maternal^6^	Residual
Pre-infection GR^1^	1392	5.12	1.31	0.35 ± 0.07	0.02	0.80 ± 0.06
Post-infection GR^1^	1359	6.85	2.82	0.21 ± 0.06	-	3.83 ± 0.24
Antibody titer^2^	1394	3.45	0.45	0.22 ± 0.05	-	0.13 ± 0.01
VL2dpi^2^	1375	4.72	1.03	0.18 ± 0.07	0.06	0.49 ± 0.03
VL6dpi^2^	1365	4.25	1.18	0.29 ± 0.06	-	0.66 ± 0.05
Viral clearance	1342	0.06	0.68	0.04 ± 0.01	-	0.41 ± 0.02

^1^Growth rate (g/day), ^2^log_10_ transformation, dpi = day post infection, VL = Viral Load ^3^Number of phenotypic records, ^4^Arithmetic mean, ^5^ Phenotypic SD from the ASREML analyses, ^6^ Variance due to dam as a proportion of phenotypic variance, Viral clearance = (Log_10_VL2dpi − Log_10_VL6dpi)/Log_10_VL2dpi.

**Table 3 genes-10-00546-t003:** Estimates (± SE) of genetic (above diagonal) and phenotypic (below diagonal) correlations based on bivariate analyses.

	Pre-Infection GR^3^	Post-Infection GR	Antibody	VL2dpi	VL6dpi	Viral Clearance
Pre-infection GR^1^		0.74 ± 0.08	0.24 ± 0.14	−0.06 ± 0.14	−0.23 ± 0.13	0.29 ± 0.20
Post-infection GR^1^	0.54 ± 0.02		0.26 ± 0.15	0.02 ± 0.17	−0.13 ± 0.16	0.15 ± 0.23
Antibody^2^	0.16 ± 0.03	0.06 ± 0.03		0.07 ± 0.17	−0.04 ± 0.14	0.30 ± 0.21
VL2dpi^2^	−0.05 ± 0.03	−0.07 ± 0.03	0.10 ± 0.03		0.17 ± 0.15	0.06 ± 0.23
VL6dpi^2^	−0.14 ± 0.03	−0.13 ± 0.03	0.03 ± 0.03	0.09 ± 0.03		−0.11 ± 0.21
Viral Clearance	0.04 ± 0.03	0.01 ± 0.03	0.04 ± 0.03	0.18 ± 0.03	−0.29 ± 0.03	

^1^ Growth rate, ^2^ log_10_ transformation, dpi = day post infection, VL = Viral Load. Viral clearance = (Log_10_Viral load, 2dpi − Log_10_Viral load, 6dpi)/Log_10_Viral load, 2dpi. ^3^Average daily gain.

**Table 4 genes-10-00546-t004:** Single nucleotide polymorphisms (SNPs) associated with NDV response traits based on genome-wise significance and positional candidate genes.

Trait	SNP	Position	*p*-Value	Candidate Genes and Location
Pre-infection_GR	AX-76523043	3:63366122	5.42 × 10^−6^	GOPC, downstream, 4697 DCBLD1, downstream, 9312 LOC421740, upstream, 596840 ROS1, upstream, 69755 UNC5D, intron LOC431251, downstream, 400465 ATP6V1B2, upstream, 242578
	AX-76262097	22:1854894	6.75 × 10^−6^	
Post-infection_GR	AX-75920682	19:1607256	3.65 × 10^−6^	AUTS2, intron SBDS, upstream, 793675 MIR1567, downstream, 99809
Antibody	AX-76035154	2:145809151	6.43 × 10^−6^	RPLP1, upstream, 3167 MIR6572, upstream, 49226 LPP, intron
	AX-77135791	9: 14444877	9.66 × 10^−6^	
Log_10_Viral load, 2dpi	AX-76811433	5:28848641	5.88 × 10^−6^	PLEKHH1, intron TMEM229B, upstream, 82089 PIGH, downstream, 22777
Log_10_Viral load, 6dpi	AX-76312211	24:429611	2.25 × 10^−9^	TIRAP, downstream, 9853 ETS1, downstream, 448924 TIRAP, downstream, 12699 ETS1, downstream, 446078 TIRAP, downstream, 4299 ETS1, downstream, 454478
	AX-76312344	24:432457	6.16 × 10^−9^	
	AX-76311970	24:424057	1.22 × 10^−8^	

**Table 5 genes-10-00546-t005:** Percentage of genetic variance explained by 1-Mb genomic regions that are associated with NDV response traits (>0.5% of genetic variance) based on the Bayes-B method.

Trait	Chr	Position Window (Mb)	#Markers	%TGV^1^
Pre-infection GR	22	1004589-1997368	509	1.15
	4	71001596-71999395	287	0.93
	11	18001466-18991342	409	0.64
	12	11001448-11994345	485	0.63
	15	4000820-4999664	625	0.59
	3	63009968-63997299	322	0.58
	20	76150-998687	317	0.51
	3	65001841-65999833	377	0.5
	2	29005343-29996746	326	0.5
	1	140114736-140998169	292	0.5
Post-infection GR	19	1000224-1999134	722	1.18
	7	28002821-28999513	472	0.55
Antibody^2^	9	13000454-13998539	492	1.08
	13	12000451-12999639	472	0.67
	14	10000304-10999961	635	0.65
	8	1000043-1999902	446	0.63
	30	48000483-48965385	175	0.56
	9	14001725-14997194	537	0.54
	10	2000005-2998892	581	0.54
Log_10_Viral load, 2dpi^2^	5	28000344-28996407	407	2
	5	41000480-41998371	353	0.8
	9	5001289-5998634	519	0.51
	7	8003124-8997158	310	0.51
Log_10_Viral load, 6dpi^2^	24	7891-999869	740	12.4
	30	21001186-21998289	341	0.71
	1	133002233-133996605	410	0.57

^1^ Percentage of total genetic variance, Traits log_10_ transformed.

**Table 6 genes-10-00546-t006:** Positions and genes located in 1 Mb windows with ≥1% of genetic variance for NDV response traits.

Trait	# SNPs	Chr: Window (Mb)	Genes
Pre-infection_GR	287	4: 71.00–72.0	PCDH7
	509	22: 1.00–2.0	TNFRSF10B, NEFM, GFRA2, NKX2-6, XPO7, NEFL, TTI2, RHOBTB2, CHMP7, ADAM28, LOXL2, NKX3-1, DOK2, DMTN, LZTS1, SLC18A1, SLC39A14, STC1, MAK16, RNF122, DUSP26, ENTPD4, SLC25A37, DOCK5, ATP6V1B2, EGR3, PBDC1, PHYHIP, C8orf58, SORBS3, NPM2, POLR3D, BIN3, PPP3CC, PEBP4, R3HCC1, LOC107050771
Post_infection_GR	722	19: 1.00–2.0	AUTS2, WBSCR17, CALN1, TYW1, MIR1587, MIR1354, MIR1567
Antibody	492	9: 13.0–14.0	UTS2B, FGF12, ATP13A4, OPA1, CCDC50, GMNC, GP5, LRRC15, OSTN, MB21D2, FCGBP, HRASLS, ATP13A5, CPN2, ATP13A3, HES1
2 dpi	407	5: 28.0–29.0	ACTN1, SLC39A9, ZFP36L1, SMOC1, SRSF5, EXD2, TMEM229B, ERH, CCDC177, RAD51B, DCAF5, GALNT16, PLEKHD1, SUSD6, MIR1710, MIR1617, SRSF5A, SLC10A1
6 dpi	740	24: 0.0–1.0	ETS1, CHEK1, H2AFX, CDON, PANX3, ST3GAL4, C2CD2L, FAM118B, STT3A, MSANTD2, SRPRA, VSIG10L2, ROBO3, RPUSD4, HYLS1, SIK2, HEPACAM, FEZ1, KIRREL3, DCPS, TIRAP, FOXRED1, PUS3, ESAM, CCDC15, SLC37A2, VPS11, HMBS, DPAGT1, PKNOX2, NRGN, MIR1758, EI24, TMEM218, ROBO4, SPA17, LOC112530272

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
