# Peer review of "Genetic Analyses of Tanzanian Local Chicken Ecotypes Challenged with Newcastle Disease Virus"

_genes, 2019, doi:10.3390/genes10070546_

Reviewer 1 Report

This paper describes the genetic analysis of Tanzania local chicken ecotypes challenged with Newcastle disease virus. The authors used anti-NDV antibody levels, viral load and growth rate to investigate the host response. Genome-wide studies revealed that five QTLs are associated with growth and response to NDV. Authors concluded that the QTLs associated with viral load at 6dpi encompasses immune related genes such as ETS1, TIRAP and KIRREL3 and might be involved in early immune response. However Saelao et al., published results failed to identify significant SNPs associated to viral load at 6dpi. Authors explained that differential heat exposure strategies might be the contributing factor for the observed differences. 

Authors presented their work in publishable format, well discussed and concluded. 

However authors need to incorporate second independent GWAS study to backup their claims.

Authors need to incorporate cell line studies to demonstrate the association of GWAS-identified genes to NDV response.

Due to contradicting report from saelao et al., authors need to clarify the differences with molecular proofs or with another independent study.       

Author Response

This paper describes the genetic analysis of Tanzania local chicken ecotypes challenged with Newcastle disease virus. The authors used anti-NDV antibody levels, viral load and growth rate to investigate the host response. Genome-wide studies revealed that five QTLs are associated with growth and response to NDV. Authors concluded that the QTLs associated with viral load at 6dpi encompasses immune related genes such as ETS1, TIRAP and KIRREL3 and might be involved in early immune response. However Saelao et al., published results failed to identify significant SNPs associated to viral load at 6dpi. Authors explained that differential heat exposure strategies might be the contributing factor for the observed differences. 

Correction: Results published by Saelao et al. identified significant SNPs associated to viral load at 6 dpi. This is in agreement with our study. Please see lines 409 – 412. The study by Saelao et al. involved heat stress and identified a similar region on chr 24 for viral load at 6 dpi. A parallel study by Rowland et al (used Hy-line brown, with no heat stress) did not identify the region on chr 24. For that reason, we speculated that the difference in results could be due to heat stress. Please see lines 414 – 420.  

Authors presented their work in publishable format, well discussed and concluded. 

Thank you

However authors need to incorporate second independent GWAS study to backup their claims.

We recognize the reviewer’s recommendation. However, the GWAS study in our manuscript is novel and presents very important results. A second GWAS study could be a future investigation but was beyond the scope of this project.   

Authors need to incorporate cell line studies to demonstrate the association of GWAS-identified genes to NDV response.

We recognize the reviewer’s recommendation. This could be a separate future project but is beyond the scope of the current project. The need for further validation of results is acknowledged in the manuscript, see lines 431 – 433

Due to contradicting report from Saelao et al., authors need to clarify the differences with molecular proofs or with another independent study.  

There is no contradiction with saelao et al. Please see lines 409 - 420. Our findings are in agreement with those of Saelao et al. for viral load at 6dpi. That is clearly discussed in the manuscript on lines 409 – 420.

 Reviewer 2 Report

The manuscript investigates genomic regions associated with immune resistance in local chickens from Tanzania challenged with Newcastle Disease Virus. 

The data is very interesting, because of the high-quality phenotypes collected and the ability to genotype all the individuals. The methodology is adequated, however, the results could be presented in a different way. The discussion could be improved, especially regarding the genes in the regions associated with the phenotypes.

Specific comments:

Introduction: 

In line 46 the authors mention that vaccination is not an adequate control. How would breeding programs be adopted by such farmers? 

Material and Methods: 

L86: It is not clear what it is meant by replicate. Is it the lab work replicate in each bird?

L 91 define days of age as doa.

L 108: there is a () after FTA cards. Maybe a citation is missing.

L 114: I would suggest removing the genotype metrics from the text, as they are stated nicely in the table. 

L 148: I believe the marker coding is -10/0/10 

L 150: There is a strange symbol in what I expected a pi. (I notice some strange symbols in other parts as well, maybe the word document or latex code should be double checked, as a ? in line 191). 

L 160: Semicolon at the end of the sentence. Is some text missing or just a typo?

Results:

I have a major comment for the result section regarding the way the associations are described. It is very confusing to see the GWAS results sometimes mentioning a window and sometimes mentioning SNP. I believe that when the term window is used is referring to BayesB results, and when SNP is being used is to GenABEL results, but it is sometimes confusing to read the results. I think the paragraph from 227 to 246 could be rewritten for clarification. 

Did the authors report p-values from BayesB, or all the p-values are from the single SNP association?

L 221: is 75,374 the PCs of the Z? Any discussion on why such high value? Maybe the diverse population and no selection?

L 227: What dies "Results of the GWAS (<= 0.5%)" mean? Is the percentage of variance explained?

Discussion:

L 305: Authors should acknowledge the high SE for the genetic correlation estimates.

I have a major comment on the discussion of GWAS results. In the paragraphs from 332 to 343 and 344 to 364, parts of the discussion seem to be missing a point. To be specific, the relationship of a gene with an effect on proliferation in the brain with the post-infection growth rate could be further explained. In a similar way the relationship with cell motility. I understand that growth rate is a very complex trait, but I don't see a clear relationship between the association and the gene function. Additionally, the relationship between Notch function and antibody level are also missing some link. Finally, on line 352 the relationship between viral load at 2dpi and the PLEKHH1 gene is not clear. 

The sentences from L344 to L 347 are confusing. They could be rewritten for clarification. 

L381: Was heat stress somehow assessed, or just presumed because of geographical location?

Author Response

The manuscript investigates genomic regions associated with immune resistance in local chickens from Tanzania challenged with Newcastle Disease Virus. 

The data is very interesting, because of the high-quality phenotypes collected and the ability to genotype all the individuals. The methodology is adequated, however, the results could be presented in a different way. The discussion could be improved, especially regarding the genes in the regions associated with the phenotypes.

Thank you. Your suggestions are addressed as shown below.

Specific comments:

Introduction: 

In line 46 the authors mention that vaccination is not an adequate control. How would breeding programs be adopted by such farmers?

The funders of the project, USAID has recently partnered with International Livestock Research Institute (ILRI). ILRI has directly worked with African small holders since 1994 and is conducting a separate study focusing at adoption of breeding programs of NDV resilient chickens.

 Material and Methods: 

L86: It is not clear what it is meant by replicate. Is it the lab work replicate in each bird?

It is not lab work replicate in each bird. Fixed, see line 136

L 91 define days of age as doa.

Fixed. See line 91

L 108: there is a () after FTA cards. Maybe a citation is missing.

Fixed. See lines 108 - 109

L 114: I would suggest removing the genotype metrics from the text, as they are stated nicely in the table. 

Fixed. See lines 115-116

L 148: I believe the marker coding is -10/0/10 

Thank you!

Fixed. See line 151

L 150: There is a strange symbol in what I expected a pi. (I notice some strange symbols in other parts as well, maybe the word document or latex code should be double checked, as a ? in line 191). 

Fixed. See line 150; line 192 (deleted)

L 160: Semicolon at the end of the sentence. Is some text missing or just a typo?

Fixed. It was a typo but has now been fixed.  Thank you!

Results:

I have a major comment for the result section regarding the way the associations are described. It is very confusing to see the GWAS results sometimes mentioning a window and sometimes mentioning SNP.

We recognize the reviewer’s concern. However, we felt it was appropriate to describe the results of the two GWAS methods (GenABEL (single-SNP) and Gensel (1-Mb genetic variance windows)) simultaneously for each trait, instead of separating it out. Any overlaps of genomic locations between the two methods were subsequently described as shown in the result section.

I believe that when the term window is used is referring to BayesB results, and when SNP is being used is to GenABEL results, but it is sometimes confusing to read the results. I think the paragraph from 227 to 246 could be rewritten for clarification. 

Thank you for pointing that out.

Fixed. See line 227-230. This paragraph should address the confusion and the subsequent text (line 231 - 247) should be clear. Because of the clarification, the order of tables 4 and 5 has been switched because of order in presentation of text.  

Did the authors report p-values from BayesB, or all the p-values are from the single SNP association?

All P values are from single SNP associations. For BayeB, we reported genetic variance explained by a 1-Megabase window. It has been fixed, see lines 229 – 230.

L 221: is 75,374 the PCs of the Z? Any discussion on why such high value? Maybe the diverse population and no selection?

You probably meant 71,374 (line 224) instead of 75374. Yes, we used that value during our multiple test correction. Please see details on lines 160 – 169 on how we determined our threshold under the multiple test correction.

 L 227: What dies "Results of the GWAS (<= 0.5%)" mean? Is the percentage of variance explained?

Fixed. See line 229

Discussion:

L 305: Authors should acknowledge the high SE for the genetic correlation estimates.

Fixed. See lines 351 – 353

I have a major comment on the discussion of GWAS results. In the paragraphs from 332 to 343 and 344 to 364, parts of the discussion seem to be missing a point. To be specific, the relationship of a gene with an effect on proliferation in the brain with the post-infection growth rate could be further explained. In a similar way the relationship with cell motility. I understand that growth rate is a very complex trait, but I don't see a clear relationship between the association and the gene function.

Thank you.

Fixed. See line368; lines 373 - 374

Additionally, the relationship between Notch function and antibody level are also missing some link.

We agree. However, we did want to note the role of HSE1 gene in immune response

Finally, on line 352 the relationship between viral load at 2dpi and the PLEKHH1 gene is not clear. 

We agree. There is no clear link between viral load at 2 dpi and PLEKHH1. However, the identified SNP associated with viral load at 2dpi was located in the intron of the gene. It is for that reason that this gene was briefly discussed and literature about it addressed.

The sentences from L344 to L 347 are confusing. They could be rewritten for clarification. 

Fixed. See lines 375 – 376.

L381: Was heat stress somehow assessed, or just presumed because of geographical location?

The heat stress was presumed because of geographical location. That is one of the possible explanation for the agreement between our results and Saelao et al. for viral load at 6 dpi and the disagreement of both studies with the parallel study of Rowland et al., which did not include heat stress. See lines 409 – 420.

 Round  2

Reviewer 1 Report

Authors have addressed all my concerns and article can be accepted for the publication.